# Ordered Mesoporous Carbon with Chitosan for Disinfection of Water via Capacitive Deionization

**DOI:** 10.3390/nano10030489

**Published:** 2020-03-09

**Authors:** Cuihui Cao, Xiaofeng Wu, Yuming Zheng, Donghai Zhang, Jianhua Chen, Yunfa Chen

**Affiliations:** 1Center for Excellence in Urban Atmospheric Environment, Institute of Urban Environment, Chinese Academy of Sciences, Xiamen 361021, China; chcao@iue.ac.cn (C.C.); ymzheng@iue.ac.cn (Y.Z.); 2University of Chinese Academy of Sciences, Beijing 100049, China; 3State Key Laboratory of Multiphase Complex Systems, Institute of Process Engineering, Chinese Academy of Sciences, Beijing 100190, China; donghaizhang71@163.com; 4Ningbo FOTILE Kitchen Ware Co. Ltd., Ningbo 315336, China; jhchen160@163.com

**Keywords:** ordered mesoporous carbon, CMK-3, capacitive deionization

## Abstract

Capacitive deionization (CDI) with water disinfection materials is a potential method to produce fresh water from aqueous solutions. Therefore, an ordered mesoporous carbon with chitosan (OMC-CS) was coated on the active carbon (AC) electrode as a capacitive deionization disinfection (CDI) electrode. Comparing with OMC-CS-4,6,8 as CDI electrodes, it was found that OMC-CS-6 as a CDI electrode had an excellent disinfection efficiency, killing about 99.99% *Escherichia coli (E. coli)* in the CDI process at an applied 1.2 V. The OMC-CS material was did not pollute the water and will not contaminate to the environment in comparison with other chemical antibacterial agents. This CDI electrode could play a huge role in biocontaminated water in the future.

## 1. Introduction

According to the 2015 Global Risk Report, which was published by the World Economic Forum (WEF), the water crisis is ranked as the top long-term risk [1]. About 783 million people do not have access to clean and safe water, and 1 in 9 people worldwide cannot obtain drinking water [2]. With the growing population and industrial development, various pollutants are discharged into the water, such as heavy metals, nitrosoamines and endocrine disrupters. These contaminants have an adverse influence on human health and the environment [3,4,5,6,7,8,9,10].

At present, the conventional methods of water treatment are as follows: coagulation, sedimentation, filtration, disinfection, decontamination, which are chemically and operationally intensive [3,5,11,12]. Furthermore, many current water disinfectant agents used for the chemical treatment of water may produce side products such as chlorine, hydrochloric acid, ammonia, alum, ozone, ferric salts, and permanganate, which are harmful to freshwater resources [3]. Therefore, there is great demand for effective, robust and low cost promising techniques for improving the water quality by decontamination and disinfection.

In addition, the selection of appropriate of antimicrobial agents for water treatment is also crucial. A new type of antimicrobial agent that kills microorganisms on contact through physical disruption of the anionic cytomembrane is a matter of great concern [8,13,14,15]. These antimicrobial agents are less resistant to microorganisms compared with the conventional antibiotics [16,17,18] and many contact-active antimicrobials are not only arbitrarily effective against both bacterial and mammalian cells, but also toxic. However, Chitosan, which has one amino group and two hydroxyl groups, is the only alkaline polysaccharose in the nature world and can be considered as a natural polyamine [19]. 

Nowadays, capacitive deionization (CDI) technology has attracted much attention due to its use of nontoxic chemicals with no contaminants, as well as being easy to operate and economical, compared with competitive technologies, including reverse osmosis (RO) [5,20] and multistage flash distillation (MSF) [21], multi-effect distillation (MED) [22] and thermal processes [23]. Therefore, it is becoming a potential technology for removing the salts in producing fresh water from saline water flowing through two opposing electrodes [24,25]. Similarly, the surfaces of bacteria are also negatively charged, they are removed through CDI process. Karthik et al. used commercially activated carbon cloth electrodes on the application of CDI for water disinfection and found that the degree of removal of bacteria (not killing) was only 82.8 ± 1.8% [16]. 

The electrosorption capacity of CDI is strongly governed by the physical properties and internal structure of the electrode materials. Therefore, the materials should not only have high porous structure, good specific surface-area electrical conductivity, specific capacitance, but also have cyclic stability and environmental friendliness [26]. Generally, the suitable candidates of the materials includes activated carbon [27,28], carbon nanotube [29,30], carbon aerogel [28,31], porous carbon [32] and graphene [33,34,35]. Ordered mesoporous carbons-CMK3 (OMCs-CMK3), which synthesized by using mesoporous silica SBA-15, sucrose, sulphuric acid as a template, a carbon source, the carbonization catalyst respectively and carbonized, have shown remarkable applications in many fields [36,37]. 

In this work, an ordered mesoporous carbon CMK-3 was synthesized through the template method. The above CMK-3 was modified by immersing in the chitosan solution, and nitrogen-containing functional groups were introduced into the surface. Then, the ordered mesoporous carbon with chitosan (OMC-CS) was coated on the activated carbon electrode as a CDI electrode for the capacitive deionization disinfection of saline water.

## 2. Experimental

### 2.1. Materials

All of the chemicals, including Pluronic P123, tetraethyl orthosilicate (TEOS), sucrose, chitosan powder and sulfuric acid were purchased from Sigma-Aldrich (Shanghai, China) and used without further purification. 

### 2.2. Preparation of CMK-3-chitosan

The mesoporous silica template (SBA-15) was synthesized through triblock copolymer Pluronic P123 as the template and tetraethyl orthosilicate (TEOS) as the silica source, respectively [38]. Simply, 4.00 g P123 was added to 144 mL 1.7 M hydrochloric acid solution and then stirred for 4 h at 313 K. Put the TEOS into the above mixture at the mass ratio of TEOS/P123 = 2 and then stirred for 2 h at room temperature. The mixture was kept into Teflon-lined sealed containers at 373 K for 48 h. The final samples were filtered, and washed with deionized (DI) water and dried for 12 h at 353 K. The mixture was placed in an oven at 573 K for 2 h. The CMK-3 was synthesized according to the reported literature [36]. Typically, 1.25 g sucrose and 0.14 g sulfuric acid were dissolved with 5 g deionized (DI) water, and stirred for 2 h. Then 1.00 g SBA-15 was added to the above solution and stirred for 4 h. The mixture was transferred to the oven at 373 K for 6 h. Then, sucrose (0.80 g), sulfuric acid (90.00 mg) and DI water (5.00 g) were added to the mixture respectively and then the above mixture was kept in the oven at 373 K for another 6 h. Next, the sample was filtered, and washed with deionized (DI) water and dried for 8 h at 353K. Finally, the sample was placed in the tube furnace at 1173 K for 6 h by a heating rate of 5 K min^−1^. After carbonization, the sample was activated in 2 mol L^−1^ NaOH solution. Then the sample was filtered and washed with DI water and ethanol and dried at 393 K. The sample (0.25 g) was modified by immersing with the prepared chitosan solution, which the chitosan powder (4, 6, 8 g) were dissolved in a 3 wt.% acetic acid respectively, at 303 K for 24 h. The samples were further immersed in 1 mol L^−1^ KOH solution overnight. After washing with DI water several times, the samples were dried at 378 K. The chitosan modified CMK-3 was denoted as OMC-CS-*x*, where *x* was the weight of chitosan.

### 2.3. Characterization

The microstructures of the samples were observed by field emission scanning electron microscopy (FEFESEM, JSM-7800) (Electronics Japan-Oxford, TKY, Japan) and transmission electron microscopy (TEM, JEM-2100) (Electronics Japan-Oxford, TKY, Japan). The Brunauer-Emmett-Teller (BET) (Anton Paar Quanta Tec Inc., Shanghai, China) and nitrogen adsorption isotherms were measured with an ASAP 2020 (Micromeritics) at 77 K. X-ray photoelectron spectroscopy (XPS) (Thermo Fisher Scientific-CN, Shanghai, China) and thermogravimetry (TG) (NSK Ltd., TKY, Japan) were measured with X-ray photoelectron spectrometer (ESCALAB 250Xi) and SDT Q600 TA.

### 2.4. CDI Electrode Fabrication

A mixture of CMK-3/OMC-CS-*x*, acetylene black and polyvinylidene fluoride at the ratio of 8:1:1 (w/w) was dissolved in a proper amount of N-Methyl pyrrolidone for cationic disinfecting CDI electrode. The above mixture was grinded thoroughly for 1 hour and then was casted onto a graphite paper. The samples were dried under vacuum at 353 K, the active electrode area was 5 × 5 cm^2^.

### 2.5. CDI Experiments

The experiments were carried out in a continuously recycling system (Figure 1). The system included a CDI cell, a peristaltic pump, a DC power supply and a conductivity monitor. The CDI cell included end plates, soft silica gel plates, electrodes and spacer. The bio-contaminated water/NaCl solution was pumped into the CDI device at a flow rate of 12 mL min^−1^. The concentration change of the solution was measured by a conductivity meter. The voltage of the working electrodes was 1.2 V.

### 2.6. Electrochemical Measurements

The electrochemical performance was evaluated by cyclic voltammetry (CV) using a three-electrode system in a CHI 660D electrochemical workstation (Shanghai CH Instruments co. LTD., Shanghai, China), which was carried out in a 0.5 M NaCl aqueous solution and in a three-compartment cell using OMC-CS-*x* electrode (the working electrode), a platinum gauze electrode (the counter electrode), and a saturated calomel electrode (the reference electrode). Here, the specific capacitances were obtained from the following equation:
$$C=\int \frac{IdV}{2v∆Vm}$$
where *C* was the specific capacitance (F/g), *I* was the response current density (A), *V* was the voltage (V), v was the potential scan rate (V/s) and m was the mass of the electrode material (g). 

### 2.7. Preparation of Microbial Cells

*Escherichia coli* (ATCC8739) was purchased from American Type Culture Collection (ATCC, Manassas, VA, USA) and broths and agar media were obtained from Becton Dickinson Company (Franklin Lakes, NJ, USA). Freeze-dried strains are first activated to obtain the corresponding bacteria, then the bacteria cells were inoculated in LB agar and cultured at 310 K overnight and then harvested, centrifuged. The cell numbers were determined by the plate colony counting method. One-hundred microliters of diluent was transferred into a culture plate with a pipette and spread with 310 K LB agar. The plates were cultured in a constant humidity thermostat at 310 K overnight for colony formation. 

### 2.8. Vitro Culture

To test the bacteriostatic properties of OMC-CS-*x* by vitro culture method, a certain number of samples were evenly dispersed with 30 min ultrasonic waves in 1 mL of sterile water. Then the samples were sterilized under the UV lamp for 30 min. One milliliter of 10^6^ CFU cells was diluted into the samples and cultured in vitro by 200 rpm shaking at 310 K. Take 0.1 mL of the samples for culturing at 310 K overnight in LB agar at 15, 30, 60, 120 and 180 min, respectively. 

### 2.9. Percent Killing Calculation of CDI

The *E. coli* was harvested by centrifugation after culturing in MHB broth for 6 h at 310 K, and then diluted to the corresponding concentration. Two milliliters of 10^8^ CFU cells was transferred into 200 mL sterile DI water to obtain 10^6^ CFU cells as the starting biocontaminated water. A sample was taken from the outlet of the CDI at an interval of 5 min for determination of cell numbers. Briefly, a 100-μL sample was transferred into the culture plates with a pipette and spread with 310 K LB agar. Then, the plates were kept in a constant humidity thermostat at 310 K overnight for colony formation. The percentage of bacteria that was killed was determined using the equation below.
$$\%\mathrm{kill}=\frac{\mathrm{cell}\;\mathrm{count}\;\mathrm{of}\;\mathrm{control}-\mathrm{survivor}\;\mathrm{count}\;\mathrm{on}\;\mathrm{sample}\;}{\mathrm{cell}\;\mathrm{count}\;\mathrm{of}\;\mathrm{control}}\times 100\%$$


## 3. Results and Discussion

### 3.1. Characterization of CMK-3

The TEM and FESEM images of CMK-3 and OMC-CS-4,6,8 are shown in Figure 2. CMK-3 and OMC-CS-4,6,8 were ordered as am arrangement of cylindrical mesoporous channels (see Figure 2a–d). However, it was found that some of the channels and surface become blurry, as shown in Figure 2b–d. The reason may be that the chitosan loaded into the pores and on the surfaces of CMK-3. The surface of CMK-3 was relatively smooth (see Figure 2e). The FESEM image of OMC-CS-4,6,8 (Figure 2f–h) indicated that chitosan did not destroy the structure of CMK-3. However, from Figure 2f–h, it could be seen that chitosan had adhered to the surface of CMK-3, the surface roughness of OMC-CS-4,6,8 had been increased. 

The impact of the chitosan loading was analyzed on the pore structure of CMK-3 by nitrogen adsorption-desorption isotherms. From Figure 3a, it was found that the isotherms of the samples could be classified as type I–type-IV according to the IUPAC. The specific surface area and total pore volume of the CMK-3 was 1291.35 m^2^/g and 1.49 cm^3^/g, respectively (Table 1), which was similar to that reported in the literature [39] of specific surface area and pore volume of 984 m^2^/g and 1.09 cm^3^/g, respectively. From Figure 3a, the surface areas and mesoporous volumes obviously decreased after modification because of the chitosan had already filled into the mesoporous channel. Figure 3b displayed that the BJH pore size distributions of samples (Table 1). The loss of pore volume was possibly associated with the introduction of chitosan onto the entrance and walls of the mesoporous carbon.

Figure 4 shows the thermogravimetric behaviors of chitosan, CMK-3 and OMC-CS-4, OMC-CS-6, and OMC-CS-8. The weight loss below 373 K was 0.1–6% duo to the removal of physiosorbed water. The weight loss of chitosan between 508 and 623 K was about 45% due to the degradation of chitosan that was similar to the result by Ge et al. [40]. For the CMK-3, the weight loss happened at 788 K under air atmosphere. The chitosan-modified CMK-3 samples (OMC-CS-4, OMC-CS-6, and OMC-CS-8) exhibited similar weight change profiles at between 508 and 623 K. According to thermogravimetric (TG) analysis, the chitosan loading on OMC-CS-4, OMC-CS-6 and OMC-CS-8 were determined as 56.28, 51.45 and 50.5 wt.%, respectively. It indicated that more chitosan of OMC-CS-4 loaded into the mesoporous channel than that of OMC-CS-6 and OMC-CS-8, while more chitosan of OMC-CS-8 was loaded on the mesoporous surface.

The elemental surface composition was determined by XPS. It was found that atomic concentration of carbon, oxygen and nitrogen based on the results of XPS. There was no nitrogen element on the CMK-3 surface. Figure 5 illustrates the XPS N 1s spectra of CMK-3 and OMC-4,6,8. Obviously, there was no nitrogen element on the CMK-3 surface; however, there was an apparent peak on the OMC-4,6,8. Therefore, it indicated that the chitosan loaded onto the CMK-3 surface successfully. From Table 2, it showed that both exist the chitosan were loaded into the mesoporous channel and on the mesoporous surface. The result was corresponded to the result of TGA.

Figure 6a presents the curves of electrical conductivity variation with time at the concentrations of NaCl solution (80 mg L^−1^). At the beginning of the experiment, the conductivity variation was fast. In the first 60 seconds of the experiment, the conductivity of NaCl solution by using CMK-3 electrode was reduced from 169.5 to 165.7 μS/cm. This reduction was significantly higher than OMC-CS-4,6,8 electrodes, which was due to the adsorption of ions was affected by the channels and surfaces of CMK-3 were loaded by chitosan. Compared with OMC-CS-4,6,8 electrodes, the conductivity of NaCl solution decreases gradually in the experiment. It indicated that with the increase of the load, the effect on ion adsorption was more obvious—as the time went on, it formed a double-layer capacitor on the surface of the electrode. The process of ion adsorption achieves dynamic equilibrium. 

After the reverse voltage was applied, the conductivity increased rapidly. The conductivity of NaCl solution by using CMK-3 electrode was increased from 164.68 to 169.5 μS/cm. This increment was higher than OMC-CS-4,6,8 electrodes. On the one hand, it was possible that the double-layer capacitor prevented the release of ions. On the other hand, the chitosan loaded in the channels and surfaces of CMK-3 also prevents the desorption of ions. With either adsorption or desorption of the ions by CMK-3 and OMC-CS-4,6,8 electrode, it did not spend too much time for that. The first process of adsorption and desorption of ions by CMK-3 and OMC-CS-4,6,8 electrode were 6.86, 11.3, 13.52 and 15.36 min, respectively. Therefore, the electrodes could absorb the ions completely by the CDI and release the ions quickly by regeneration.

Figure 6b,c presents the effluent NaCl concentration/time profiles and electrosorptive capacity/time profiles of CMK-3 and OMC-CS-4,6,8 electrodes. It was found that the result corresponded to the conductivity/time profiles of CMK-3 and OMC-CS-4,6,8 at NaCl 80 mg L^−1^. It was possibly related to the double-layer capacitor and the chitosan loaded in the channels and surfaces of electrode.

The CV curves of the CMK-3 and OMC-CS-4,6,8 electrodes at different scan rates (5, 10, 15, 25, 50, 100 mV s^−1^) are presented in Figure 7a–h. From Figure 7, it can be seen that the curves are irregular/rectangular. It indicated that Na^+^ and Cl^−^ ions diffusion increases with voltage increase. While the voltage was up to 0.3 V, a wide reduction peak appeared in Figure 7c–h. It indicated that the concentration of electrolyte tends to zero and the *I_pc_* reaches the maximum. However, the result of the CV curves of OMC-CS-4,6,8 electrodes was different from the CMK-3 electrode. It indicated that after loading the chitosan on the CMK-3, it would not only affect the diffusion of ions, but also the magnitude of current. On the other hand, the current intensity was increasing as the scan rate increased because the performance of ion movement would be changed by the change of scanning rate. From Figure 7d, it was found that while the scan rate was 100 mV/s, the reduction peak appeared in 0.02 V, it was possible that the performance of the ions diffusion after loading the chitosan on the CMK-3. 

### 3.2. Antimicrobial Activity of OMC-CS-4,6,8

Chitosan derivatives such as QC and GO have been reported to have low haemolytic activity and low toxicity to mammalian cells [41,42]. The *E. coli* (Gram-negative bacterium) was chosen as model pathogens for testing the antimicrobial activity. Figure 8a–c shows the results of vitro culture of *E. coli* with OMC-CS-4,6,8 at 15, 30, 60, 120 and 180 min, respectively. From Figure 8d, it was found that the respective % kills for the OMC-CS-4,6,8 at 15 min were 65.85%, 63.41% and 51.22%. The respective % kills for the OMC-CS-4,6,8 at 60 min were 82.93%, 63.41% and 75.61%. The respective % kills for the OMC-CS-4,6,8 at 180 min were 92.68%, 95.12% and 90.24%. At the beginning of vitro culture, the % kills for the OMC-CS-4 was higher than OMC-CS-6,8, but with the increase of time, % kills for the OMC-CS-6 was the highest. The % kills for the OMC-CS-6 was from 63.41 to 95.12%, the efficiency was the best. It indicated that the amount of chitosan loaded on CMK-3 was different antimicrobial activity. 

From the above results, we studied the killing mechanism of CMK-3 and OMC-CS-*x* in the vitro culture suspension. FESEM of the morphological changes of *E. coli* after treatment with CMK-3 and OMC-CS-*x* solution (100 μg mL^−1^) indicated that *E. coli* was killed by CMK-3/OMC-CS-*x* through contact-active physical disruption (Figure 9). Compared with the smooth surfaces of untreated control cells (Figure 9a), the surfaces of the *E. coli* cells treated with CMK-3 (Figure 9b) became wrinkled. On the other hand, it was found that the *E. coli* cells treated with OMC-CS-*x* not only have physical defects at the two ends of the cell, which was able to be caused by the edges of OMC-CS-*x*, but also have distinct areas of damage on the cells, and the cell envelopes appeared severely collapsed (Figure 9c–e). Thus, it indicated that the OMC-CS-*x* effectively disrupted the membranes of *E. coli* to cell death.

### 3.3. CDI Process 

The report found that the graphene oxide-graft-quaternized chitosan nanohybrid electrode was coated for CDI had the high-performance capacitive deionization disinfection of water [43]. The OMC-CS-4,6,8 electrodes were applied for CDI of microbes in biocontaminated sterile water. *E. coli* were mostly absorbed into the surface of OMC-CS-*x* electrode under the action of the electric field, and then the sterile water outflowed into the CDI cell. In addition, the CDI static culture of *E. coli* with OMC-CS-4,6,8 electrodes without applied voltage was used as the control. Figure 10a–c showed that the results of controls at 15, 30, 60, 120 and 180 min, respectively. From Figure 10d, the respective percentage of kills for the OMC-CS-4 electrode at 15–180 min are from 36.58% to 73.17%. The respective percentage of kills for the OMC-CS-6 electrode at 15–180 min are from 51.21% to 80.48%. The respective % kills for the OMC-CS-8 electrode at 15–180 min are from 48.78% to 48.04%. It found that the static sterilizing effect of CDI without applied voltage was not very significant because the *E. coli* could not be rapidly enriched on the electrodes in a short time without applied voltage. Therefore, it was difficult to exert the effect of CDI. 

Figure 11a–c showed that the results of CDI of *E. coli* with OMC-CS-4,6,8 electrodes at 5, 10, 15, 20, 25 and 30 min, respectively. From Figure 11d, the respective percentage of kills for the OMC-CS-4 electrode at 5–30 min were from 58.54% to 99.99%. The respective percentage of kills for the OMC-CS-6 electrode at 5–30 min were from 70.73% to 99.99%. The respective percentage of kills for the OMC-CS-8 electrode at 5–30 min were from 73.17% to 97.56%. It found that the percentage of kills for the OMC-CS-6 electrode at 25 min was 99.99%, the efficiency was higher than OMC-CS-4,8 electrode. The % kills for the OMC-CS-6 electrode at 15 min was up to 92.68%. Therefore, based on the above dates, OMC-CS-6 CDI electrodes have an excellent % kill of *E. coli*. 

From the above results, we studied the killing mechanism of CMK-3 and OMC-CS-*x* in the CDI process. Figure 12 a showed that *E. coli* was enriched on the surface of the disinfecting OMC-CS-*x* electrode after the disinfection step. The surface of dead *E. coli* was wrinkled (Figure 12a), which corroborated that OMC-CS-*x* physically damaged the *E. coli* envelope and disrupted the membranes of *E. coli* to cell death. After the voltage was cut off in the regeneration process, *E. coli* on the electrode surface were washed away (Figure 12b), which showed that the OMC-CS-*x* electrode had an excellent regeneration and recyclability.

OMC-CS-*x* material combined the synergistic effects of nanoscale thickness, mesoporous size, electrical conductivity of CMK-3, and bactericidal properties of chitosan. Therefore, OMC-CS-*x* material showed different functions, such as CMK-3 with chitosan improved the dispersion of CMK-3 to increase the aggregation tendencies. The anionic *E. coli* bacterial cell would be enriched by the cationic OMC-CS-*x* due to modification of CMK-3 with chitosan inverted the charge state of CMK-3 from anionic to cationic to cause *E. coli* bacterial cell membrane disruption and death.

## 4. Conclusions

A unique disinfecting OMC-CS-*x* material was synthesized and explored as part of a novel CDI electrode. The OMC-CS-6 electrode can achieve ultra-high killing (99.99% after 25 min) of 10^6^ CFU mL^−1^
*E. coli* through the CDI cell. During the CDI process, there was no observable OMC-CS contamination to the water. This disinfection strategy is better than all traditional methods of water disinfection and has distinct characteristics: no observable electrode materials was left out of the CDI, and the OMC-CS-6 electrode contact killing process was ultrafast, cyclically stable, with low energy consumption. Therefore, this CDI process with the OMC-CS-6 electrode, which has the advantages of strong bactericidal ability, rapid reaction, no pollution and energy savings, and has a good application prospect in the fields of biomedicine, environment and personal care.

## Figures and Tables

**Figure 1 nanomaterials-10-00489-f001:**
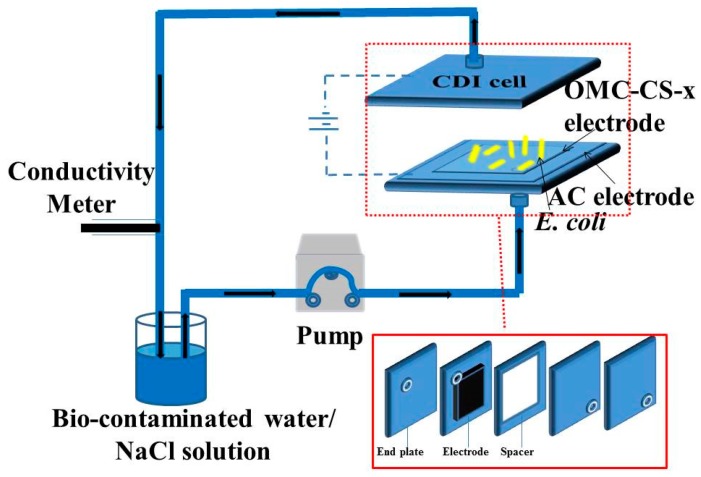
Schematic diagram of the CDI device.

**Figure 2 nanomaterials-10-00489-f002:**
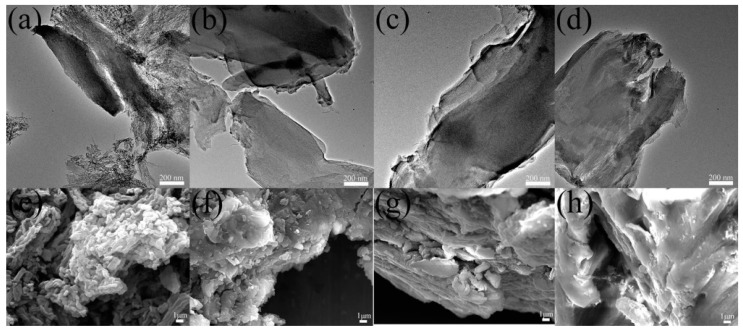
TEM and FESEM image of CMK-3 and OMC-CS-4,6,8: (**a**,**e**) CMK-3, (**b**,**f**) OMC-CS-4, (**c**,**g**) OMC-CS-6, and (**d**,**h**) OMC-CS-8.

**Figure 3 nanomaterials-10-00489-f003:**
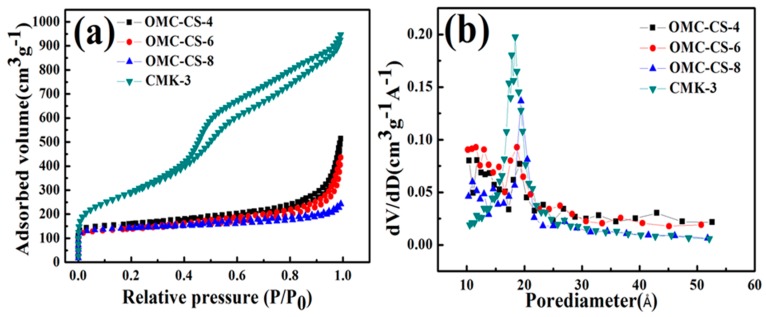
(**a**) Nitrogen adsorption-desorption isotherms at 77 K of CMK-3 and OMC-4,6,8, (**b**) Pore size distribution of CMK-3 and OMC-4,6,8.

**Figure 4 nanomaterials-10-00489-f004:**
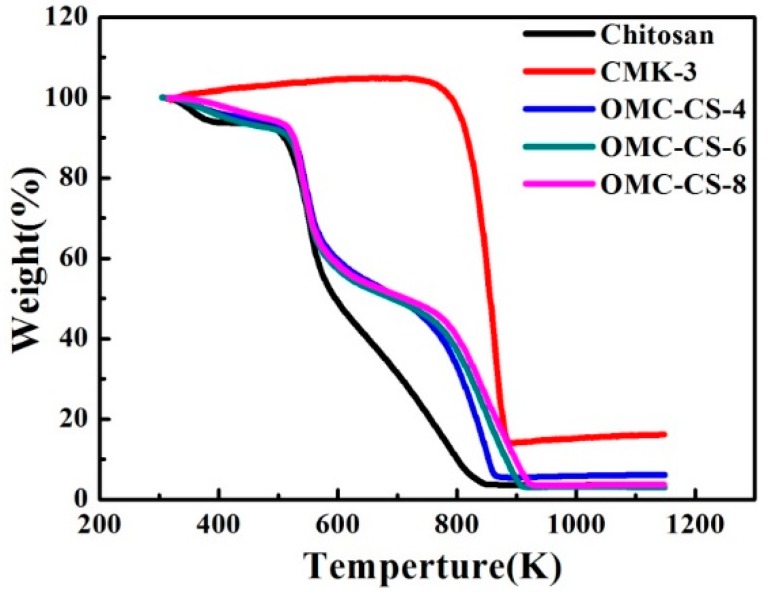
Thermogravimetric weight change curves under air atmosphere for chitosan, CMK-3, and OMC-4,6,8.

**Figure 5 nanomaterials-10-00489-f005:**
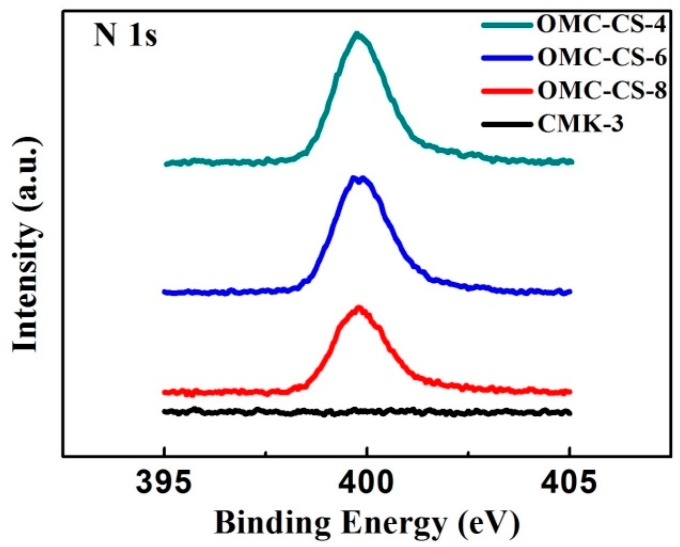
N 1s spectra of CMK-3 and OMC-4,6,8.

**Figure 6 nanomaterials-10-00489-f006:**
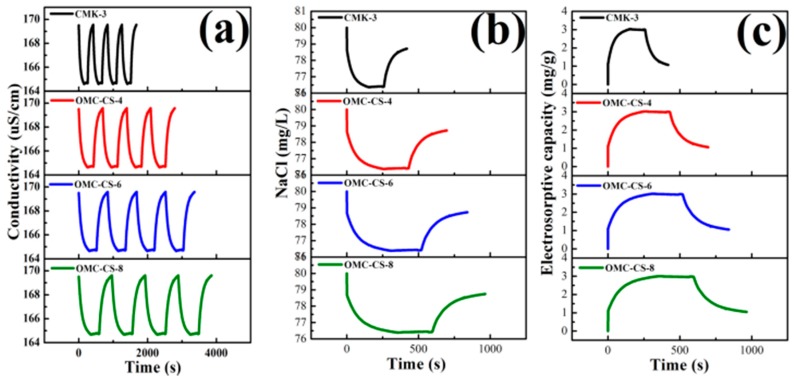
(**a**) Conductivity/time profiles of CMK-3 and OMC-CS-4,6,8 at NaCl 80 mg L^−1^; (**b**) Effluent NaCl concentration/time profiles of CMK-3 and OMC-CS-4,6,8 at 1.2 V; (**c**) Electrosorptive capacity/time profiles of CMK-3 and OMC-CS-4,6,8 at 1.2 V.

**Figure 7 nanomaterials-10-00489-f007:**
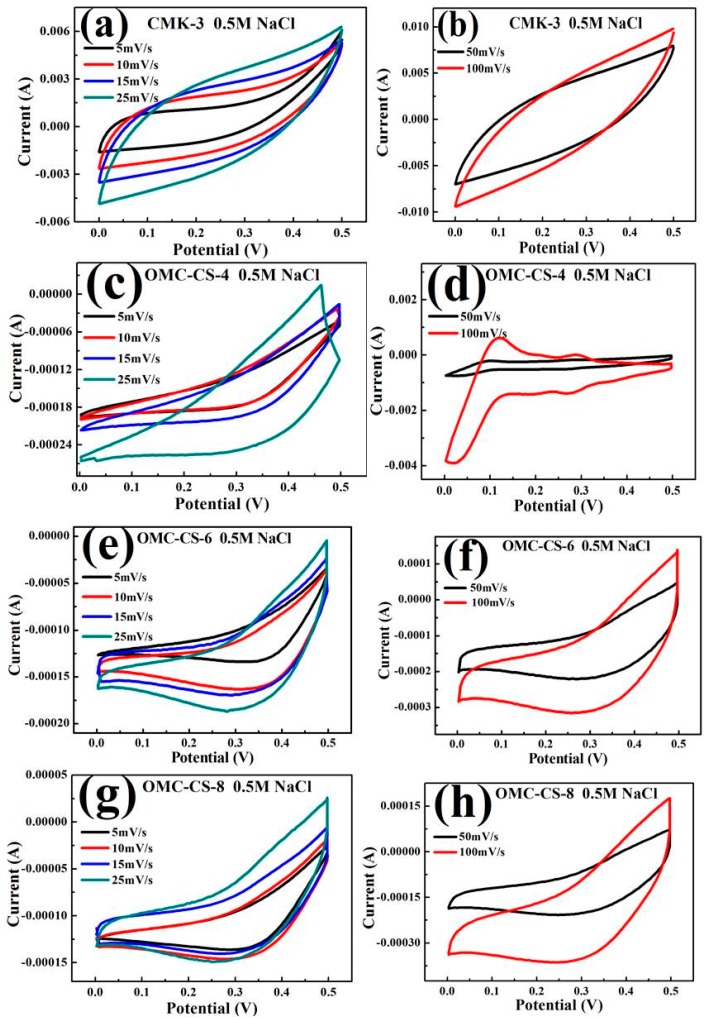
(**a**,**b**) Cyclic voltammograms of CMK-3; (**c**,**d**) Cyclic voltammograms of OMC-CS-4; (**e**,**f**) Cyclic voltammograms of OMC-CS-6; (**g**,**h**) Cyclic voltammograms of OMC-CS-8 at different scan rates.

**Figure 8 nanomaterials-10-00489-f008:**
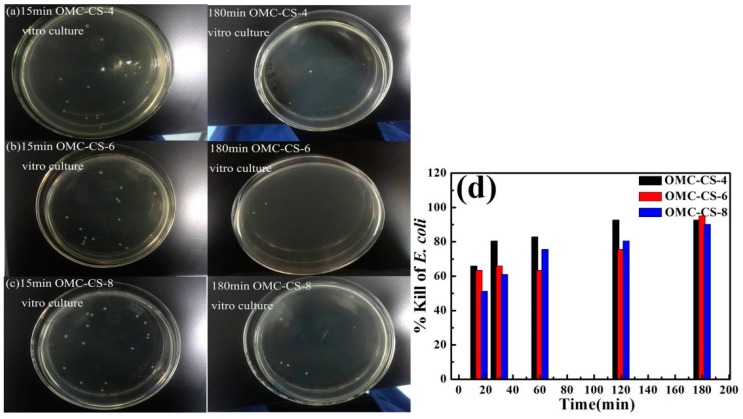
(**a–c**) The colony forming units; (**d**) Killing rate of microbes after vitro culture with OMC-CS-4,6,8 (100 μg mL^−1^) for 15, 30, 60, 120, 180 min at 10^6^ CFU mL^−1^.

**Figure 9 nanomaterials-10-00489-f009:**
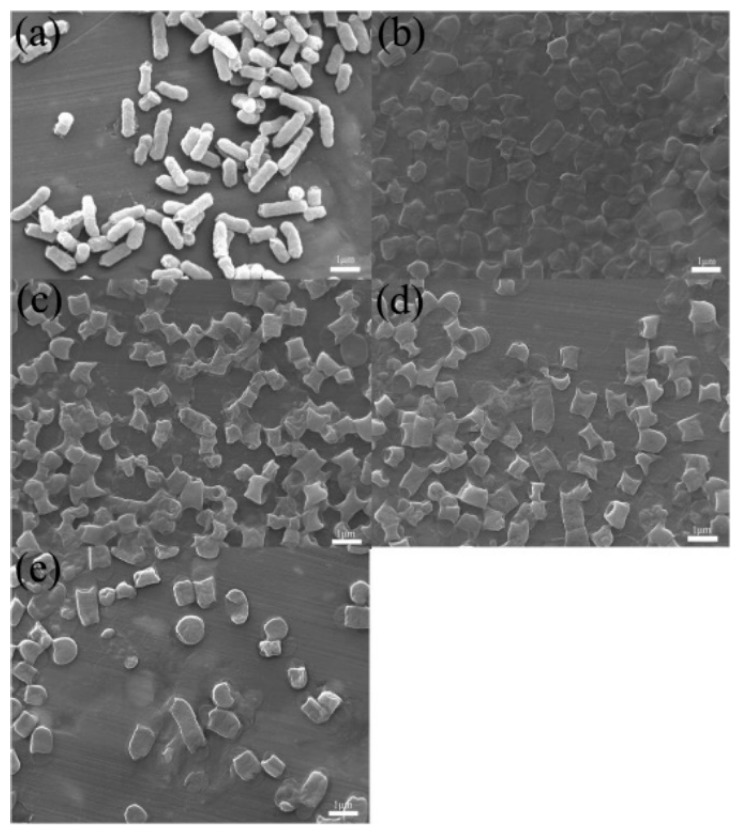
FESEM image of *E. coli* untreated control (**a**), treated with CMK-3 (**b**) and OMC-CS-4,6,8 (**c–e**) (100 μg mL^−1^) for 60 min.

**Figure 10 nanomaterials-10-00489-f010:**
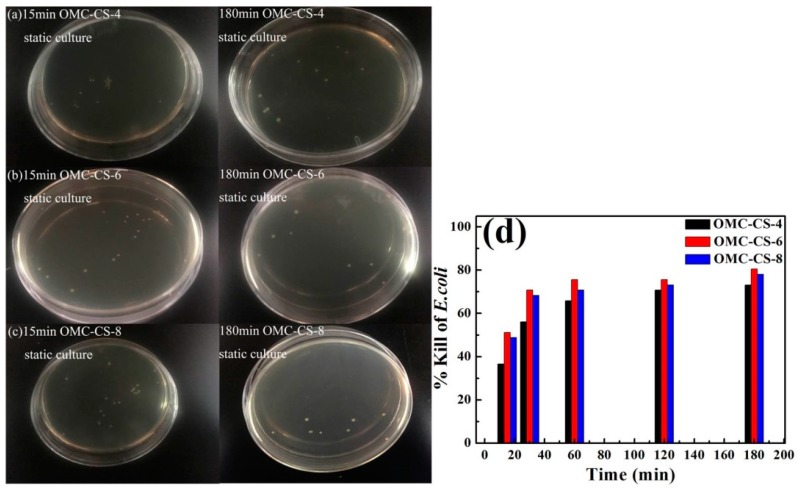
(**a–c**) The colony forming units; (**d**) Killing rate of microbes; after CDI static culture with OMC-CS-4,6,8 (100 μg mL^−1^) for 15, 30, 60, 120, 180 min at 10^6^ CFU mL^−1^.

**Figure 11 nanomaterials-10-00489-f011:**
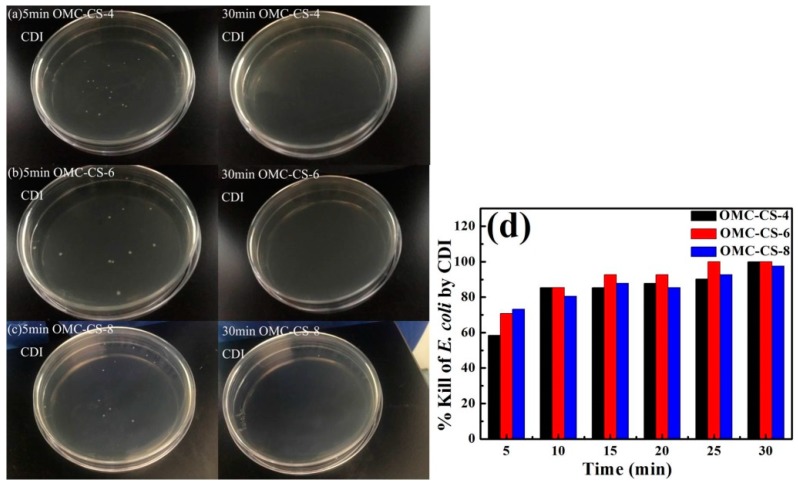
(**a**–**c**) The colony forming units; (**d**) Killing rate of microbes; after CDI process with OMC-CS-4,6,8 electrodes for 5, 10, 15, 20, 25, 30 min at 10^6^ CFU mL^−1^.

**Figure 12 nanomaterials-10-00489-f012:**
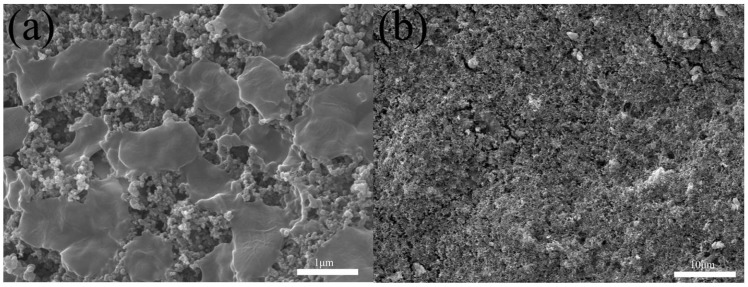
FESEM images of the OMC-CS electrode after the CDI process (**a**) and the regeneration process (**b**).

**Table 1 nanomaterials-10-00489-t001:** Comparison data from adsorption isotherms.

Sample	*S_BET_* (m^2^/g)	*V_tot_* (cm^3^/g)	*D_pore_* (Å)
CMK-3	1291.35	1.49	23.2
OMC-CS-4	193.87	0.59	61.9
OMC-CS-6	193.39	0.50	51.8
OMC-CS-8	119.09	0.19	31.2

**Table 2 nanomaterials-10-00489-t002:** Surface elemental composition of CMK-3 carbons and OMC-CS-4,6,8 from XPS N 1s spectra.

Sample	Atomic Concentration (%)		
	C	O	N
CMK-3	94.91	5.09	0.00
OMC-CS-4	65.62	28.70	5.68
OMC-CS-6	59.88	33.08	7.04
OMC-CS-8	58.68	34.08	7.25

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
