# Peer review of "Ordered Mesoporous Carbon with Chitosan for Disinfection of Water via Capacitive Deionization"

_nanomaterials, 2020, doi:10.3390/nano10030489_

Round 1

Reviewer 1 Report

In this study, the authors introduced an ordered mesoporous carbon with chitosan as a CDI electrode for water disinfection. Overall, this is an interesting idea however the current version of the manuscript has the following major issues and thus I recommend re-evaluation of the manuscript after the major revision:

My biggest concern is why the authors investigates the electrochemical performance with a NaCl solution if the main focus is disinfection (Figs. 6 and 7). What is the relation between the desalination and disinfection processes? What did we learn from the desalination experiments (Figs. 6 and 7) that’s relevant to the main objective of this study and required for the validation of the hypothesis regarding disinfection? I believe a better story is needed regarding the research design and presentation. Another important issue is the motivation and significance of this study were not clearly stated. Why OMC-CMK3 is selected? Why is it modified to OMC-CS? What makes OMC-CS an ideal candidate for this application compared with other available materials. A better discussion/introduction is needed for CDI as a disinfection method. What other methods can be used? What is the state-of-the-art performance and issues in terms of water disinfection? The abstract and the conclusion require moderate English corrections - several sentences need to be re-written (E.g., “Comparing with OMC-CS-4,6,8 as CDI electrodes, it found that…”. These sections should also be expanded to tell more about the motivation and significance of this study. Other minor issues: Line 33 – what does chemically intensive mean? Line 36, a space is needed – resources [3] Line 45 – the sentence starting with “The allure of capacitive deionization….” is too long and difficult to read. Paragraph 3 in the Introduction seems out of place – a better transition between paragraphs 2,3 and 4 would be useful. Line 59 – What does CMK3 stand for? The fonts of sub-labels (i.e., a, b, c, d) of the figures can be improved for consistency. Huge fonts are used for some of the figures, such as Figs. 7, 8d and 9d, while it’s really difficult to read the others, such as Figs. 8a, 8b, 8c.  I would probably include Fig. 3 before Table 1 and Fig. 5 before Table 2 as the figures provide the required info for the tables. Throughout the manuscript, an English editing for minor issues/spell checks is required. 

Reviewer 2 Report

The manuscript by Chen and co-workers is surely interesting and the quality of the data is generally high.

One question to start with: is the adjective "macro" in the title a deliberate choice or a typo? The paper deals clearly with mesoporous, not macro, materials.

The role played by a surface electric field in the antimicrobial activity is interesting. The authors claim that E.coli cells were adsorbed (there is a spelling error, this is adsorption, not absorption) on (and not into) the surface of the OMC-CS-x electrode under the action of electric field. Is that claim that migration under a field is responsible for the segregation of cells at the surface? Or is perhaps as I would be inclined to think that there are modification of the adsorptive properties of the cell membrane in response to an electrostatic stimuls?

Maybe testing the revesibility of the adsorption (or cell morphology) wold help in defining the relative importance of altered mass transport vs altered cell surface chemistry.

I think that it will be beyond the scope of this paper to clarify on the chemical changes to the cell surface upon an electrical stimuls, but perhaps it would be a nice addition to the current discussion (and add scope to further investigations) to highlight the role that a near surface electric field can have on bond forming and bond breking processes (e.g. Shaik, S.; Ramanan, R.; Danovich, D.; Mandal, D., Structure and Reactivity/Selectivity Control by Oriented-External Electric Fields. Chem. Soc. Rev. 2018, 47, 5125–5145;  Zhang, L.; Laborda, E.; Darwish, N.; Noble, B. B.; Tyrell, J.; Pluczyk, S.; Brun, A. P. L.; Wallace, G. G.; Gonzalez, J.; Coote, M. L.; Ciampi, S., Electrochemical and Electrostatic Cleavage of Alkoxyamines. J. Am. Chem. Soc. 2018, 140, 766–774; (1) Aragonès, A. C.; Haworth, N. L.; Darwish, N.; Ciampi, S.; Bloomfield, N. J.; Wallace, G. G.; Diez-Perez, I.; Coote, M. L., Electrostatic catalysis of a Diels–Alder reaction. Nature 2016, 531, 88)

Author Response

This manuscript is a resubmission of an earlier submission. The following is a list of the peer review reports and author responses from that submission.